# Is the Magnesium Content in Food Supplements Consistent with the Manufacturers’ Declarations?

**DOI:** 10.3390/nu13103416

**Published:** 2021-09-28

**Authors:** Anna Puścion-Jakubik, Natalia Bartosiewicz, Katarzyna Socha

**Affiliations:** Department of Bromatology, Faculty of Pharmacy with the Division of Laboratory Medicine, Medical University of Białystok, Mickiewicza 2D Street, 15-222 Białystok, Poland; natalia_bartosiewicz@wp.pl (N.B.); katarzyna.socha@umb.edu.pl (K.S.)

**Keywords:** magnesium, pharmacy, food supplements, drugstore

## Abstract

Food supplements (FS) are gaining more and more popularity because they are a quick way to compensate for deficiencies in the diet. Due to their affordable price and easy-to-take form, they are eaten by all age groups and by healthy and sick people. There are many categories of this type of preparations on the market, and FS with magnesium (Mg) are some of the most commonly used. Therefore, the aim of the study was to determine the Mg content in FS and to compare the estimated value with that declared by the manufacturer. The study included 116 FS containing Mg. In order to determine the Mg content, the atomic absorption spectrometry (AAS) method was used. The tested FS were divided in terms of the declared content, pharmaceutical form, chemical form of Mg, composition complexity, and price. It was shown that in the case of 58.7% of the samples, the Mg content was different than the permissible tolerance limits set by the Polish chief sanitary inspectorate, which range from −20% to +45%. It has been estimated that as a result of the differences in the content, the patient may take up to 304% more Mg per day or 98% less than it is stated in the declaration. The above results indicate that the quality and safety of FS should be more closely monitored.

## 1. Introduction

Food supplements (FS) are preparations that are intended to supplement the diet with deficient substances. They contain minerals, vitamins, and other substances that can cause a specific physiological effect, such as fatty acids, amino acids, or probiotics. They come in various pharmaceutical forms, including tablets, capsules, powders, and ampoules. They do not contain medicinal substances, so they cannot be used to treat disease entities [1].

Minerals and vitamins that may be used in FS in Poland are listed in the Regulation of the Minister of Health of 17 September 2018. The minimum amount of vitamins and minerals in a daily portion in FS should not be less than 15% of the reference consumption values. However, maximum acceptable levels are established on the basis of the upper safe levels of consumption, the amounts provided in the diet, and the recommended intake for the population. This value must be safe for the health and life of the consumer [2].

FS are very common, and their popularity and market share are steadily increasing. Data from 2019 indicate that more than half of adults, both in Europe and the US, use FS [3]. According to research by Li et al. (2010), these are more often women than men (38.6% vs. 28.5%), as well as people over 50 (57.4%) and with higher education [4]. This is a consequence of the increased demand for nutrients in these groups. Despite the fact that FS cannot replace a balanced diet, their intake allows them to supplement existing deficiencies. In the elderly, it is particularly important because physiological changes in the body with age and long-term use of many medications make seniors most exposed to nutrient deficiencies.

Our previous research has shown that the use of FS is very widespread. Supplements containing magnesium (Mg) are used by approximately 8% of medical university students [5]. Other data from Poland indicate that FS with Mg account for 7.56% of the market. Preparations with this ingredient are very popular in Poland—about 25.0% of respondents use them. Among the inhabitants of Spain, 13.4% of men take them and among the inhabitants of Germany 18.3% of men and 20.4% of women [6].

The legal regulations governing the FS market in Poland are both national and European requirements. An important legal act is the Food and Nutrition Safety Act of 25 August 2006 [1], Regulation of the Minister of Health of 18 May 2010 amending the regulation on the composition and labeling of FS [7] and the Regulation of the Minister of Health of 17 September 2018, on the composition and labeling of FS [2]. The Regulation of the Minister of Health of 17 September 2018 [2] lists vitamins and minerals and their chemical forms that may be present in supplements. For substances authorized to appear in supplements, which were listed in the above regulation [2], further purity criteria are defined. Dyes and additives should meet the purity requirements specified in Commission Regulation (EU) No 231/2012 of 9 March 2012 [8]. Despite the above-mentioned legal acts being in force, the registration procedures and placing the FS on the market are very simple and only require presentation documentation and packaging design to the chief sanitary inspector. No quality or safety tests are required, which creates a risk that there are low-quality preparations on the market that differ in terms of their composition from the manufacturers’ declarations.

Mg is necessary, inter alia, to maintain normal cell function, muscle contraction, including the heart muscle, and conditions nervous excitability [9,10]. Mg deficiency has also been shown to contribute to the development of oxidative stress in obese people [11], disturbances in mineral homeostasis such as Mg may interfere with cancer progression [12], and Mg supplementation may play a beneficial role in controlling asthma by acting as an anti-inflammatory and bronchodilator [13].

The causes of Mg deficiency include: reduced gastrointestinal absorption, loss of Mg from gastrointestinal tract, increased renal loss, excessive sweating, increased requirements (for example in pregnancy), or older age, which disrupts many processes [14].

The above factors prompted us to evaluate the Mg content in FS available on the Polish market, coming from local producers, but also producers known in various European countries. To our knowledge, this is the first study that covers such data as a variety of chemical forms, pharmaceutical forms or preparations at different prices.

## 2. Materials and Methods

### 2.1. Materials

Samples of dietary supplements were selected on the basis of previously conducted surveys [5] and on the basis of popularity in the largest chain pharmacies in the country.

Inclusion criteria included: popularity among patients, availability category ‘FS’, preparations within the expiry date. 

The following exclusion criteria were adopted: occasional sales, over-the-counter ‘OTC’ availability category.

The study included 116 FS purchased in stationary pharmacies as well as online. In order to assess the quality of FS in the best possible way, preparations were selected for the research, which differed in terms of composition, pharmaceutical form, price and, manufacturer. Detailed characteristics of the studied FS are presented in Appendix A.

FS were taken from three different blisters or as three subsamples, analyzed in triplicate (statistically insignificant differences between the determinations) were harvested and tested in 2020–2021.

### 2.2. Sample Preparation

FS were ground in a vibrating grinder (Testchem, Poland) and weighed into Teflon mineralization vessels of about 0.3 g with an accuracy of 1 mg (exact weights were recorded). Then 4 mL of spectrally pure concentrated (69%) nitric acid (Tracepur, Merck, Darmstadt, Germany) were added. The microwave mineralization process was carried out in a closed system (Berghof, Speedwave, Eningen, Germany), according to the following program:Step 1: 170 °C, 10 min, 20 atm., 80% of microwave power;Step 2: 190 °C, 10 min, 30 atm., 90% of microwave power;Step 3: 210 °C, 10 min, 40 atm., 90% of microwave power;Step 4: 50 °C, 18 min, 40 atm., 0% of microwave power.

The obtained mineralizates were quantitatively transferred to polypropylene vessels with deionized water.

### 2.3. Determination of Mg Content

Mg content was determined by atomic absorption spectrometry (AAS), acetylene-air flame technique with Zeeman background correction. The determination was carried out using the Z-2000 instrument (Hitachi, Tokyo, Japan). Before the analysis, all of the analyzed samples were diluted, depending on the declared content of the tested element. Lanthanum chloride (1% LaCl_3_, Sigma-Aldrich, Merck, Darmstadt, Germany) was used as the sequestering agent. The assay was performed at a wavelength of 285.2 nm and 7.5 mA current lamps. The concentration was read from the curve prepared using a 1 mg/mL Mg standard solution for AAS (Merck, Germany). The limit of detection (LOD) and limit of quantification (LOQ) were 0.26 mg/kg and 0.78 mg/kg, respectively.

The conducted research did not require the approval of the Bioethics Committee of the Medical University of Bialystok.

### 2.4. Validation of Method

In order to control the accuracy of the analyses, a certified reference material was used (Simulated Died D, LIVSMEDELS VERKET, National Food Administration, Uppsala, Sweden). The determination was performed before the analysis and after each 10 determinations. All values were within the certified value range (2740–3100 mg/kg). The accuracy (% of error) was 0.67%, and the coefficient of variation V = 1.57%.

### 2.5. Comparison of Results with the Standards Adopted by the Chief Sanitary Inspectorate in Poland

In accordance with the guidelines adopted by the European Commission in 2012 on establishing tolerance limits for minerals contained on labels, the obtained values were compared with the guidelines adopted by the Commission, amounting to −20 to +45% for FS-containing minerals [15,16].

### 2.6. Statistical Analyses

Statistica software (Tibco, Palo-Alto, CA, USA) was used for calculations and statistical analyzes. The results are presented as mean (Av.) with standard deviation (SD), minimum (Min), maximum (Max), as well as median (Med.), and lower quartile (Q1), upper quartile (Q3), interquartile range (IQR). 

## 3. Results

The results of the analyses are presented in Table 1, Table 2, Table 3, Table 4 and Table 5. The following classification criteria were used: declared content, pharmaceutical form, chemical form of Mg, amount of minerals (only Mg or multi-component preparations), and price.

In the case of preparations with a declared content below 100 mg of Mg per portion, the highest determined content was 202.0 mg, and the lowest was only 1.5 mg (Table 1).

The second criterion of the division was the criterion of the pharmaceutical form. The formulation with the highest reported content (795.7 mg/portion) was available as a powder to be dissolved in water. The lowest marked values were for supplements available in the form of capsules (1.5 mg/portion), effervescent tablets (4.9 mg/portion), and tablets (5.8 mg/portion).

Most of the studied FS contained Mg in the form of Mg citrate (*n* = 35) and Mg carbonate (*n* = 34), while the least contained in the form of hydroxide (*n* = 2) and glycerophosphate (*n* = 1). Both the preparation with the lowest determined content of Mg (1.5 mg/portion) and the preparation with the highest determined content (795.7 mg/portion) contained Mg citrate (Table 3).

Out of 116 FS tested, 75 contained only Mg among the minerals. This category includes both the preparation with the lowest determined value (1.5 mg) and the preparation with the highest determined mg content (795.7 mg) (Table 4).

Interestingly, in preparations with a lower price (below PLN 10), the highest mean Mg content was recorded at the level of 192.1 ± 191.1 (Table 5).

The chief sanitary inspectorate, responsible for the quality of FS, allows the deviation of minerals from −20% to +45%. Figure 1 shows the variation in individual samples. It was shown that 58.7% of FS were outside the acceptable range (Figure 1a,b).

At a further stage, we also assessed by how many percent the expected value by consumers would differ from the actual value consumed, in accordance with the manufacturer’s recommendation, because the tested FS can be taken in amounts greater than just one portion per day. It has been shown that for 54.1% of FS, consumers will consume a lower amount of Mg. For example, for 3.4% it will be 90–100% less than the expected value. Worryingly, in the case of one of the studied FS, consumers, using one portion of the FS each day, will consume as much as 300% more Mg than indicated on the packaging (Figure 2).

There were no statistically significant differences (*p* > 0.05) between the previously discussed factors (declaration of Mg content, pharmaceutical form, chemical form, composition, price) and the percentage of samples within the norm, below and above the norm (Table 6).

## 4. Discussion

FS are foodstuffs taken by patients and consumers to supplement existing deficiencies. Their use is not intended to treat or prevent diseases in humans, unlike drugs, and are not required to be subject to detailed qualitative and quantitative research prior to sale, unlike medicinal products. Moreover, their side effects are not monitored rigorously. This generates the need to test their quality. Our research covered more than 100 FS, which may reflect the assortment of the largest pharmacy chains.

As part of the study conducted by the SW RESEARCH agency (2017), a survey was conducted among 807 adults. It was estimated that 72.4% of Poles use FS, and about half of them systematically, i.e., 48%. Worrying is the fact that only 17% consult a doctor or pharmacist before starting supplementation. The most common reasons for taking these preparations were the desire to strengthen the body (55.4%), increase resistance to infections (44.3%), and supplement the daily diet with the missing ingredients (40.7%). Alarmingly, 6% of people argued taking supplements is the current fashion. The respondents declared that during the purchase they were guided by the composition (41.7%), price (38%), their own experience (36.6%), the recommendation of a doctor or pharmacist (34.5%), opinions of other people (25.9%), scientific certificates (16.9%) and other factors (27.7%). Satisfactory is the fact that the most frequent place of purchase was the pharmacy (65%). According to 54.9% of people, taking supplements brought them noticeable benefits, while 41.6% did not notice an effect on their health, while 3.4% were dissatisfied with the effects. In this study, more frequent use of supplements by women (51.7%) than men (48.3%) was observed, as well as among people with higher education (45.5%) [17].

Mg, next to potassium, is the most important intracellular cation. It activates over 300 enzymes. It participates, among others, in neuromuscular conduction, regulation of the body’s mineral homeostasis, regulation of blood pressure, insulin metabolism, and muscle contractility. It is a macronutrient necessary for proper functioning, therefore it should be supplied with a balanced diet. A number of factors, including the consumption of highly processed food, contribute to its reduced amount in the diet [18].

It is disturbing that if one of the tested FS is consumed, the patient will be take in 300% more Mg every day than it is stated in the declaration on the packaging. Taking too much of a dose may have side effects. There is no evidence that food-derived Mg can have a negative effect on the body, while in the case of excessive consumption of Mg from various types of supplements or medicinal products, cases of harmful effects have been reported. Since Mg salts are laxative when used in large amounts, osmotic diarrhea may occur. Symptoms also include difficulty breathing, sleep disturbances, changes in heartbeat, muscle weakness and confusion. In extreme cases, when it is accompanied by impaired renal function, serious neurological symptoms may occur, such as, among others, increased axonal excitability threshold, paralysis of the striated and cardiological muscles, including inhibition of heart contractions or prolongation of the QT interval [19]. Serious side effects, including death, were found after children took 2400 mg of Mg, which is three servings of the supplement with the highest content determined in this study [20]. The maximum amount of Mg allowed in FS is 400 mg/day [21], while our study showed almost twice as much Mg in one of the preparations.

The subject of comparing the declared and determined Mg content has so far been rarely discussed in scientific publications. Literature data indicate that the interest of researchers in FS containing Mg is also focused on assessing the safety of their use, due to the presence of potential contaminants [22,23].

It seems necessary to conduct patient education and large-scale campaigns. A 2014 study showed that every fourth Pole was unable to correctly define the definition of a FS. Almost half of the people, as many as 41%, claimed that these products have healing properties, while 31% of people assessed that they were synonymous with vitamins, and 8% that they were synonymous with minerals. Moreover, half of the respondents (50%) believed that they were subject to the same control as drugs [24].

The presence of a large amount of FS on the market makes their control very difficult. As a result of the easy registration process, more and more of them appear on the market. Data from Poland show that from 2007 to 2017, over 29,000 were entered into the register of products in the FS category. In 2016, the chief sanitary inspector received about 600 notifications about the introduction of a new preparation on the market every month [6]. This indicates the need to introduce greater restrictions, preventing the placing on the market of preparation of inappropriate quality.

Ensuring an adequate level of Mg in the body is essential for its proper functioning. The benefits of using it have been reported even in diseases of various pathogenesis, e.g., ulceration [25]. As a result of taking FS of inadequate quality, containing several times the lower doses of Mg than declared by the manufacturer, supplementing the deficiencies may be ineffective, which may result in the lack of the effects expected by patients.

The absorption of Mg is greater from food than from supplements [26,27,28,29]. Therefore it is necessary to properly balance the diet so that taking these preparations will not be necessary. However, when it is impossible or when there is an increased demand for this element as a result of other factors, supplementation with high-quality preparations should be used [30,31]. A diet deficient in terms of Mg is quite common. For example, a 2009 study assessed eating habits among people living in Russia, the Czech Republic and Poland. Consumption in line with the recommendations of the standards was shown only in about 65% of the respondents. The highest consumption of this element was among Poles, who consumed 286 mg. Czechs supplied 278 mg and Russians only 97 mg [32].

Such a large discrepancy between the declared values and those actually marked is very surprising. It may result from the improper production process of FS, lack of final product control, and inadequate labeling of the supplement. FS are sold in grocery stores, drugstores, herbal, and medical stores and pharmacies. It seems necessary that FS sold in pharmacies should be subject to greater control, which will improve their quality and increase consumer confidence in this category of food products, sold in a form analogous to drugs.

The limitations of this study are as follows: due to the heterogeneity of FS quality, other batches may have a different Mg content, the study describes the most popular preparations on the market, although the quality of FS produced by niche producers may be different. The limitation is also the large market share of preparations in one pharmaceutical and chemical form. Further research should be based on the assessment of the bioavailability of various Mg compounds and the actual concentration obtained in the blood of patients after FS ingestion.

## 5. Conclusions

The assessment of the quality of food containing magnesium showed that the declared and actual values in most dietary supplements differed. Only in two samples of the supplements were they identical. Only 41.3% of the tested samples were within the acceptable range of deviations for minerals, in line with the recommendations of the chief sanitary inspectorate. During the intake of food supplements covered by this study by patients, as a result of differences between the declared value and the measured value, the amount of the consumed element may change in the amount from a maximum of 98% less to 304% more than the declared value. Food supplements should be routinely monitored to improve their quality.

## Figures and Tables

**Figure 1 nutrients-13-03416-f001:**
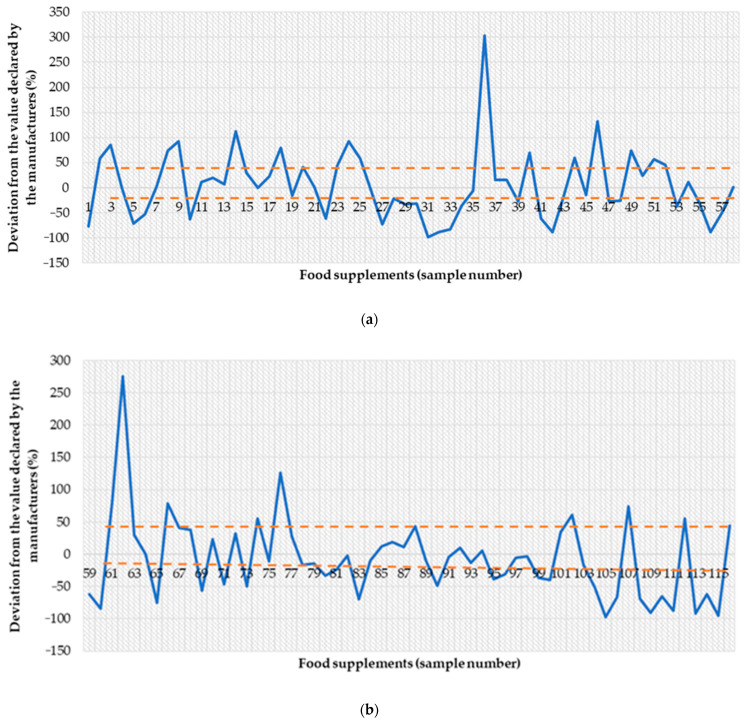
The discrepancy between the declared magnesium content and that determined in dietary supplements (**a**) samples of food supplements from 1 to 58, (**b**) samples of food supplements from 59 to 116.

**Figure 2 nutrients-13-03416-f002:**
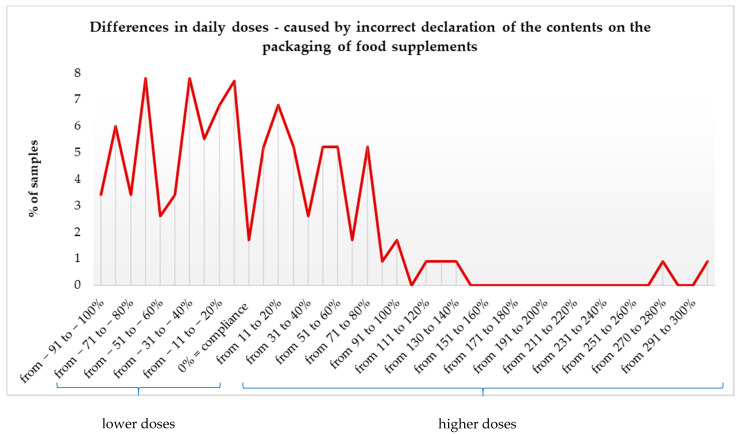
The percentage of food supplements and the difference in the actually taken magnesium dose, resulting from an incorrect declaration of the content.

**Table 1 nutrients-13-03416-t001:** Magnesium content (mg/portion) in food supplements depending on the declared magnesium content.

Declared Content	*n*	Mg Content (mg/Portion)					
		Av. ± SD	Min–Max	Med.	Q1	Q3	IQR
Less than 100 mg	49	49.7 ± 38.0	1.5–202.0	40.7	23.7	73.4	49.7
100–200 mg	45	144.9 ± 109.5	7.4–469.6	115.7	66.3	207.2	141.0
Above 200 mg	22	387.0 ± 200.2	39.1–795.7	348.7	249.2	479.2	230.0

Av.—average, IQR—interquartile range, Max—maximum value, Med.—median, Min—minimum value, Q1—lower quartile, Q3—upper quartile, SD—standard deviation.

**Table 2 nutrients-13-03416-t002:** Magnesium content (mg/portion) in food supplements depending on the pharmaceutical form.

Form	*n*	Mg Content (mg/Portion)					
		Av. ± SD	Min–Max	Med.	Q1	Q3	IQR
Capsules	13	103.8 ± 110.0	1.5–298.5	69.2	22.8	193.0	170.2
Coated tablets	11	68.5 ± 60.4	19.0–202.0	48.0	23.7	77.5	53.8
Dragees	2	78.3 ± 22.7	62.2–94.3	78.3	62.2	94.3	32.1
Effervescent tablets	24	231.2 ± 196.0	4.9–696.9	168.2	78.6	364.2	285.6
Granulates	1	233.1 ± 0.0	-	-	-	-	-
Jelly beans	1	27.5 ± 0.0	-	-	-	-	-
Liquids	7	198.4 ± 120.6	34.0–360.1	219.7	75.4	317.6	242.2
Powders	12	264.2 ± 247.2	22.1–795.7	189.1	81.2	367.4	286.2
Tablets	45	106.8± 133.6	5.8–696.5	60.2	31.4	120.8	89.4

Av.—average, IQR—interquartile range, Max—maximum value, Med.—median, Min—minimum value, Q1—lower quartile, Q3—upper quartile, SD—standard deviation.

**Table 3 nutrients-13-03416-t003:** Magnesium content (mg/portion) in food supplements depending on the chemical form.

Chemical Form	*n*	Mg Content (mg/Portion)					
		Av. ± SD	Min–Max	Med.	Q1	Q3	IQR
Magnesium bisglycinate	6	161.4 ± 103.1	28.5–317.6	154.3	93.8	219.7	126.0
Magnesium carbonate	34	132.2 ± 164.2	5.8–696.9	73.9	40.7	137.8	97.1
Magnesium citrate	35	168.4 ± 201.1	1.5–795.7	79.1	31.5	232.0	200.6
Magnesium glycerophosphate	1	78.8 ± 0.0	-	-	-	-	-
Magnesium hydroxide	2	215.6 ± 263.8	29.1–402.1	215.6	29.1	402.1	373.0
Magnesium lactate	11	45.7 ± 39.4	1.8–129.3	35.4	7.4	77.9	70.5
Magnesium oxide	8	207.6 ± 155.5	18.8–449.4	225.6	52.9	317.9	265.1
Several chemical forms	19	181.2 ± 148.1	22.8–479.2	145.4	61.2	267.6	206.4

Av.—average, IQR—interquartile range, Max—maximum value, Med.—median, Min—minimum value, Q1—lower quartile, Q3—upper quartile, SD—standard deviation.

**Table 4 nutrients-13-03416-t004:** Magnesium content (mg/portion) in food supplements depending on the amount of minerals.

Amount of Minerals	*n*	Mg Content (mg/Portion)					
		Av. ± SD	Min–Max	Med.	Q1	Q3	IQR
Only magnesium (or vitamin B6)	75	164.8 ± 183.6	1.5–795.7	93.8	34.0	249.2	215.3
Multicomponent preparations	41	124.7 ± 125.6	4.8–469.6	76.4	38.5	188.8	150.2

Av.—average, IQR—interquartile range, Max—maximum value, Med.—median, Min—minimum value, Q1—lower quartile, Q3—upper quartile, SD—standard deviation.

**Table 5 nutrients-13-03416-t005:** Magnesium content in food supplements depending on the price.

Price (PLN)	*n*	Mg Content (mg/Portion)					
		Av. ± SD	Min–Max	Med.	Q1	Q3	IQR
<10	41	192.1 ± 191.1	13.3–696.9	108.3	35.4	317.6	282.1
10–20	57	112.0 ± 138.8	1.5–795.7	74.4	30.5	129.3	98.7
>20	18	178.4 ± 164.2	22.8–649.8	113.1	71.5	267.6	196.1

Av.—average, IQR—interquartile range, Max—maximum value, Med.—median, Min—minimum value, PLN—currency in force in Poland, Q1—lower quartile, Q3—upper quartile, SD—standard deviation.

**Table 6 nutrients-13-03416-t006:** Relationship between factors and percentage of food supplements with normal, below and above normal magnesium levels (*p* > 0.05).

Criterion	Subgroups	*n*	Below Standard *n* = 46 (%)	Norm *n* = 48 (%)	Above Normal *n* = 22 (%)
Declared content	Less than 100 mg	49	19 (16.4)	24 (20.7)	6 (5.2)
	100–200 mg	45	20 (17.2)	15 (12.9)	10 (8.6)
	Above 200 mg	22	7 (6.0)	9 (7.6)	6 (5.2)
Form	Capsules	13	9 (7.8)	3 (2.6)	1 (0.9)
	Coated tablets	11	5 (4.3)	5 (4.3)	1 (0.9)
	Dragees	2	0 (0.0)	2 (1.7)	0 (0.0)
	Effervescent tablets	24	5 (4.3)	12 (10.3)	7 (6.0)
	Granulates	1	0 (0.0)	0 (0.0)	1 (0.9)
	Jelly beans	1	0 (0.0)	1 (0.9)	0 (0.0)
	Liquids	7	3 (2.6)	3 (2.6)	1 (0.9)
	Powders	12	3 (2.6)	4 (3.4)	5 (4.3)
	Tablets	45	21 (18.1)	18 (15.5)	6 (5.2)
Chemical form	Magnesium bisglycinate	6	2 (1.7)	3 (2.6)	1 (0.9)
	Magnesium carbonate	34	11 (9.5)	15 (12.9)	8 (6.9)
	Magnesium citrate	35	16 (13.8)	12 (10.3)	7 (6.0)
	Magnesium glycerophosphate	1	0 (0.0)	1 (0.9)	0 (0.0)
	Magnesium hydroxide	2	1 (0.9)	0 (0.0)	1 (0.9)
	Magnesium lactate	11	6 (5.2)	5 (4.3)	0 (0.0)
	Magnesium oxide	8	3 (2.6)	4 (3.4)	1 (0.9)
	Several chemical forms	19	7 (6.0)	8 (6.9)	4 (3.4)
Amount of minerals	Only magnesium (or vitamin B6)	75	35 (30.2)	25 (21.6))	15 (12.9
	Multicomponent preparations	41	11 (9.5)	23 (19.8)	7 (6.0)
Price (PLN)	<10	41	16 (13.8)	17 (14.7)	8 (6.9)
	10–20	57	25 (21.6)	22 (18.9)	10 (8.6)
	>20	18	5 (4.3)	9 (7.8)	4 (3.4)

## Data Availability

Data are available from the authors.

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
