# Peer review of "Is the Magnesium Content in Food Supplements Consistent with the Manufacturers’ Declarations?"

_nutrients, 2021, doi:10.3390/nu13103416_

Round 1
Reviewer 1 Report
In this manuscript the authors analyzed the magnesium content in dietary supplements marketed in Poland and compared the measured value with that declared by the manufacturer, finding that in most cases the magnesium content is significantly higher or lower than the declared data. These data are interesting, but the manuscript requires a minor revision.
Line 176: I suggest to replace “They do not have medicinal properties” with “Their use is not intended to treat or prevent diseases in humans”, as EFSA stated.
Line 263 “The absorption of Mg is greater from food than from supplements”: the sentence needs reference.
Some new dietary supplements use liposomial or sucrosomial magnesium. It would be interesting to know the measured magnesium content compared to what is claimed. Do the authors have data on this?
Reviewer 2 Report
The manuscript titled “Is the Magnesium Content in Dietary Supplements Consistent With The Manufacturers' Declarations?” addresses the monitoring of Mg content in a survery of Food Supplements with specific declared contents of the mineral commonly sold in Poland. Overall, the study is fits well with the aims and scope of the Journal. It is well written and organized. References are enough to support the entire work and the findings may be of great help to regulatory agencies. For these reason, I recommend a minor revision. Comments follow:
2.1. Materials
-in what time range was the study conducted?
-I’d like to see a list of the supplements arranged at least in declared content of Mg, categories, and the number of FS for each of them
2.2 Sample preparation
What do you mean for “wet microwave mineralization”?
2.4. Validation of method
Line 120: “analyses” please be consistent throughout the manuscript
What about LOD and LOQ of Mg?
